# The Clinical Utility of Leukoaraiosis as a Prognostic Indicator in Ischemic Stroke Patients

**Foteini Christidi** [1], **Dimitrios Tsiptsios** [1,*], **Anastasia Sousanidou** [1], **Stefanos Karamanidis** [1], **Sofia Kitmeridou** [1], **Stella Karatzetzou** [1], **Souzana Aitsidou** [1], **Konstantinos Tsamakis** [2], **Evlampia A. Psatha** [3], **Efstratios Karavasilis** [4], **Christos Kokkotis** [5], **Nikolaos Aggelousis** [5] and **Konstantinos Vadikolias** [1]

1. Neurology Department, School of Medicine, Democritus University of Thrace, 68100 Alexandroupolis, Greece
2. Institute of Psychiatry, Psychology and Neuroscience (IoPPN), King's College London, London SE5 8AF, UK
3. Department of Radiology, School of Medicine, Democritus University of Thrace, 68100 Alexandroupolis, Greece
4. Medical Physics Laboratory, School of Medicine, Democritus University of Thrace, 68100 Alexandroupolis, Greece
5. Department of Physical Education and Sport Science, Democritus University of Thrace, 69100 Komotini, Greece
* Correspondence: tsiptsios.dimitrios@yahoo.gr

**Abstract:** Stroke constitutes a major cause of functional disability with increasing prevalence among adult individuals. Thus, it is of great importance for both clinicians and stroke survivors to be provided with a timely and accurate prognostication of functional outcome. A great number of biomarkers capable of yielding useful information regarding stroke patients' recovery propensity have been evaluated so far with leukoaraiosis being among them. Literature research of two databases (MEDLINE and Scopus) was conducted to identify all relevant studies published between 1 January 2012 and 25 June 2022 that dealt with the clinical utility of a current leukoaraiosis as a prognostic indicator following stroke. Only full-text articles published in English language were included. Forty-nine articles have been traced and are included in the present review. Our findings highlight the prognostic value of leukoaraiosis in an acute stroke setting. The assessment of leukoaraiosis with visual rating scales in CT/MRI imaging appears to be able to reliably provide important insight into the recovery potential of stroke survivors, thus significantly enhancing stroke management. Yielding additional information regarding both short- and long-term functional outcome, motor recovery capacity, hemorrhagic transformation, as well as early neurological deterioration following stroke, leukoaraiosis may serve as a valuable prognostic marker poststroke. Thus, leukoaraiosis represents a powerful prognostic tool, the clinical implementation of which is expected to significantly facilitate the individualized management of stroke patients.

**Keywords:** ischemic stroke; leukoaraiosis; white matter hyperintensities; rehabilitation; prognosis; recovery

## 1. Introduction

Despite the significant progress that has been made towards early diagnosis and therapeutic interventions, ischemic stroke (IS) still ranks second among the leading causes of death in the world behind only heart disease, whereas it constitutes the main source of acquired disability in adults [1,2]. Taking into account both the disease's age-related character, with more than 50% of patients being over the age of 65 [3], and the constantly extending lifespan with an expected tripling of the individuals aged over 60 in developed countries by 2050 [4], stroke survivors' numbers are presumed to rise tremendously. Thus, a crucial need emerges for prompt and accurate identification of patients with an unfavorable prognosis that may enable to plan and implement a personalized rehabilitation, further oriented towards each individual's recovery propensity [5].

Stroke's heterogeneity in terms of etiology and pathophysiology is being reflected in the quite challenging outcome prognosis poststroke, leading to a growing interest in establishing valid stroke recovery biomarkers [6,7]. Various clinical assessment tools with a prognostic potential have been developed and utilized in an acute stroke setting [8], including the National Institutes of Health Stroke Scale (NIHSS) [9], while a number of neurophysiological techniques have been implemented [10] to provide important insight into each patient's recovery capacity following stroke and guide personalized stroke care.

Up to today, several biomarkers have been investigated to reliably forecast patients' functional outcome and recovery potential poststroke [11,12], with leukoaraiosis (LA) being among them. LA, also known as white matter lesions (WMLs), stands for a neuroimaging phenomenon frequently observed among elderly individuals and refers to specific abnormalities of the white matter (WM), typically present as either multifocal or diffuse changes of varying sizes and predominantly located within the periventricular space [13]. The aforementioned lesioned areas were first described in 1987 by Hachinski and colleagues as bilateral and symmetrical WM changes (WMCs) in the proximity of the cerebral ventricles or within the semioval center found on brain scans of older or demented subjects [14]. Depending on the neuroimaging method, LA is being recognized as hypodense brain areas on computed tomography (CT) scans or WM hyperintensities (WMHs) on T2-weighted (T2WI) and/or fluid-attenuated inversion recovery sequences (FLAIR) magnetic resonance imaging (MRI) sequences [15].

As far as the epidemiological profile of WMLs is concerned, LA burden seems to be significantly associated with advanced age, with substantially increasing prevalence among patients over 60 years old. More specifically, although half of healthy individuals aged between 44 and 48 years presented with some degree of WMCs on brain imaging, WM abnormalities are prominent in approximately 95% of elderly subjects aged 60 to 90 years [16–18]. Furthermore, the observation of WMLs within a younger healthy population appears to be a rather uncommon finding, while age greater than 55 years is found to be coupled with a 10-fold increase in the prevalence of WMCs when compared to age ≤55 years [19].

Apart from the chronological age, both hypertension (HT) and diabetes mellitus (DM) are considered two major risk factors for LA development, since high blood pressure levels and elevated glucose levels are associated with WMLs accumulation, thus highlighting the substantial influence of known vascular risk factors on the early onset and late progression process of WM abnormalities [20,21]. Similarly, tobacco use and high serum homocysteine levels may contribute to the LA occurrence, while alcohol consumption and dyslipidemia with increased plasma low-density lipoprotein-cholesterol (LDL-C) levels seem to play a role in the LA progression, but not the disease onset [13,22–24].

Regarding the pathophysiological basis of LA concept, although the exact mechanism remains only poorly understood, the development of WM abnormalities is considered to be of ischemic origin. The accumulation of small vessel disease (SVD) risk factors in healthy subjects and the subsequent lipohyalinosis of small vessels significantly enhance the burden of vascular injury to the aging brain by contributing to the impairment of cerebral autoregulation, venous hypertension, blood–brain barrier (BBB) disruption and activating a cascade of pathophysiological events leading to cerebral hypoperfusion [25]. As a result of the aforementioned hemodynamic alterations, WM ischemia occurs being accompanied by a substantially decreased cerebral blood flow within the lesioned brain areas. Reflecting chronic and diffuse ischemic injury, LA is characterized by the presence of specific neuropathologic features, including axonal loss, myelin degeneration, gliosis, enlargement of periventricular spaces. The role of hypoperfusion as an underlying etiology of WMCs accumulation is supported by a growing body of literature [26].

Providing further insight into the neurochemical mechanisms underlying LA pathogenesis, the assessment of brain metabolites N-acetyl aspartate (NAA), choline (Cho) and creatine (Cr), that mirror neuronal integrity, membrane synthesis and degradation and energy metabolism, respectively, may further elucidate the role of each individual's neuro-

chemical profile on the WMCs accumulation. Patients presented with LA are commonly characterized by abnormal metabolic WM alterations compared to healthy subjects, with significantly decreased both NAA/Cho and NAA/Cr ratios in lesioned WM areas, thus indicating neuronal loss or dysfunction among patients with LA [27].

Along with lacunar infarctions, LA constitutes an imaging marker of SVD, a degenerative vascular disease most commonly observed among elderly individuals. Although WMLs were considered as incidental brain imaging findings with absent clinical impact, accumulated evidence suggest that LA burden is coupled with a group of specific clinical manifestations, including gait impairment with subsequent falls, urinary dysfunction, disability, cognitive decline, as well as mood disorders. Gait and balance disturbance accompanied by frequent falls within an elderly population is strongly associated with the severity of detected WMCs [28]. Regarding urinary disturbances, the degree of WM abnormalities is significantly correlated to urinary urgence, since individuals presented with severe LA report substantially higher incidence of urinary urgence compared to elderly subjects with only mild or moderate LA on brain imaging scans [29].

As far as the linkage between WMLs and the impairment of cognitive function is concerned, LA burden is associated both with an accelerated decline in global cognitive performance and a negative influence on specific cognitive domains including executive functions, attention and processing speed [18,30]. Interestingly, WMCs may yield additive information regarding the cognitive profile of non-disabled aged individuals, acting as an independent predictor of cognitive decline [31,32]. Alongside with cognitive decline, increasing WMLs volume can forecast the development of mild cognitive impairment, dementia and disability [18]. More specifically, WMCs in conjunction with lacunar infarcts and cerebral microbleeds (CMBs) are considered to be the major underlying pathology in cases of vascular dementia [33]. With respect to the relationship between LA severity and psychiatric disorders, progression of WM abnormalities was found to be causally linked to the development of depressive symptoms, highlighting the crucial role of LA in the pathogenesis of late-life depression [34].

According to Leukoaraiosis and Disability (LADIS) study, the role of LA burden as an independent predictor of transition from an autonomous functional status to disability is of key importance. Evaluating the impact of WMCs on functional performance of elderly subjects reveals a greater than 2-fold higher transition risk towards disability among aged individuals with severe LA compared to those with only mild or moderate LA [35].

Several assessment tools have been utilized to quantify the presence and severity of WMLs on CT or MRI brain scans, including visual rating scales and volume measurement methods. The most commonly implemented approach is the Fazekas score which evaluates both periventricular and deep WMCs in terms of number and size of the lesions and differentiates between focal or punctate, early confluent and confluent abnormalities [36]. Apart from that, CMBs, visible as hypointense signals on MRI, seem to share a common development pathway with LA. Interestingly, WMCs severity was found to be paralleled with CMBs presence, with both being strongly associated with the clinical manifestation of cerebrovascular symptoms [37,38].

Taking into consideration the clinical relevance and the potential prognostic role of baseline LA within an aging population, as well as the emerging need for accurately forecasting each stroke individual's propensity for recovery, the purpose of the present study was to review all available literature published within the last decade dealing with LA as an outcome predictor in an acute ischemic stroke (AIS) setting.

## 2. Materials and Methods

The Preferred Reporting Items for Systematic Reviews and Meta-analyses (PRISMA registration number: CRD42022364723) was used to guide this study. Our study's methods were a priori designed.

## 2.1. Search Strategy

Two investigators (AS and DT) conducted literature research of two databases (MEDLINE and Scopus) to trace all relevant studies published between 1 January 2012, and 25 June 2022. Search terms were as follows: ("leukoaraiosis" OR "white matter hyperintensities" OR "WMHs") AND ("ischemic stroke" OR "brain infarction") AND ("prognosis" OR "recovery" OR "outcome"). The retrieved articles were also hand searched for any further potential eligible articles. Any disagreement regarding the screening, or selection process, was solved by a third investigator (KV), until a consensus was reached. Figure 1 presents the review flowchart.

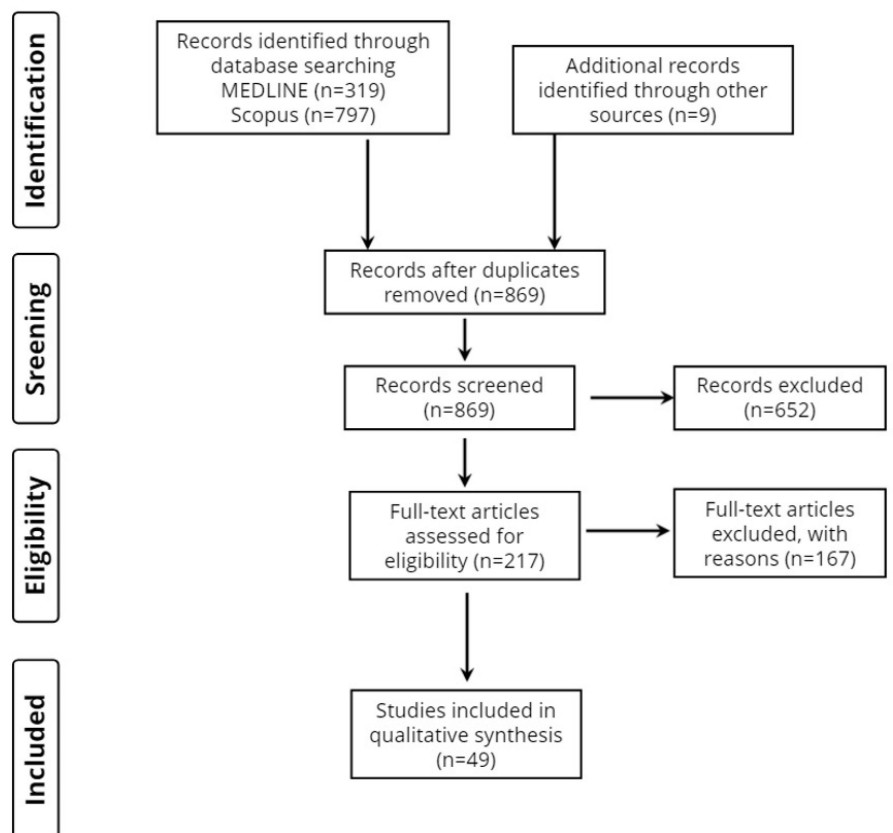

**Figure 1.** Study flow chart (PRISMA diagram).

## 2.2. Selection Criteria

Only full-text original articles published in the English language were included. Secondary analyses, reviews, guidelines, meeting summaries, comments, unpublished abstracts, or studies conducted on animals were excluded. There was no restriction on study design or sample characteristics.

## 2.3. Data Extraction

Data extraction was performed using a predefined data form created in Excel. We recorded (when available) authors, year of publication, the type of stroke, the type of study, number of participants and their age (including mean/median, standard deviation, range when available), gender, level of education, marital/occupational status, body mass index (BMI), cerebrovascular risk factors, medication, previous stroke, follow-up time, method of LA assessment, time of MRI, the scale used to assess stroke severity and clinical outcome and main findings.

*2.4. Data Analysis*

No statistical analysis or meta-analysis was performed due to the high heterogeneity among studies. Thus, the data were only descriptively analyzed.

## 3. Results

*3.1. Database Searches*

Overall, 1125 records were retrieved from the database search. Duplicates and irrelevant studies were excluded; hence, a total of 217 articles were selected. After screening the full text of the articles, 49 studies were eligible for inclusion.

*3.2. Study Characteristics*

Forty-nine publications fulfilled our inclusion criteria, as shown in Table 1. Forty-two studies focused entirely on AIS, 2 studies included patients with either AIS or TIA and 5 studies included patients regardless of stroke type. Considering the origin of the studies, 25 studies were from Asia, 12 studies were from America, 10 studies were from Europe and 2 studies were from Africa.

*3.3. Time of Clinical Outcome*

Two studies focused on hemorrhagic transformation, 9 studies on early neurological deterioration (END) or early neurological improvement (ENI) within 1-week post-AIS, 10 studies on early stroke outcome (<1 month), 18 studies on short-term outcome (1–3 months), 9 studies on long-term outcome (3–12 months), 3 studies on chronic outcome (>1 year), whereas 3 studies focused on clinical outcome upon discharge from a rehabilitation facility.

*3.4. Method of Leukoaraiosis Neuroimaging Assessment*

Thirty-two studies preferred the Fazekas score, 4 studies used the van Swieten scale, 3 studies applied the Age-Related White Matter Change Scale, 1 study used the Cardiovascular Health Study rating scale, 1 study used the semiquantitative visual rating system described by Scheltens and colleagues, 1 study used the leukoaraiosis score previously described by Reid and colleagues, 1 study firstly described a scoring system, 1 study differentiated between the presence or absence of leukoaraiosis and 11 studies estimated the WMH volume on MRI.

*3.5. Study Design*

In total, most of the studies included in this review were longitudinal and they were either retrospective or prospective cohorts. Only 2 studies were cross-sectional [39,40].

**Table 1.** Characteristics of the 49 included studies.

| 1st Author, (Year) | Type of Stroke, Study Design, Participants (*n*) | Demographics: Age (Years), Gender (M/F), Education (Years or Level), Marital/Occupational Status, Income, BMI | Cerebrovascular Risk Factors (*n*) | Medication (*n*) | Previous Stroke (*n*) | Follow up Time | Leukoaraiosis/ WMH Assessment | Time of MRI | Scale of Stroke Severity and Prognosis/Clinical Outcome | Main Findings |
|---|---|---|---|---|---|---|---|---|---|---|
| *Hemorrhagic transformation* | | | | | | | | | | |
| Wei, (2017) [41] | Cardioembolic stroke due to atrial fibrillation or rheumatic heart disease, Longitudinal, 251 | 68.49 ± 12.43, 99M/152F | Hypertension (*n* = 105), Diabetes mellitus (*n* = 27), Hyperlipidemia (*n* = 3), Alcohol (*n* = 43), Smoking (*n* = 44) | Anticoagulants (*n* = 26), Antiplatelets (*n* = 32), Statins (*n* = 12), | 51 | Until discharge | Fazekas score, Van Swieten scale and ARWMC scale | Within 5 days after admission | NIHSS and GCS score on admission | Higher median NIHSS score, lower median GCS score, larger infarct areas, lower levels of serum total cholesterol or low-density lipoprotein on admission, periventricular and frontal moderate to severe LA are associated with higher HT risk. |
| Wang, (2022) [42] | Acute Cerebellar Infarction, Longitudinal, 190 | 61.84 ± 12.16, 141M/49F | Hypertension (*n* = 149), Diabetes mellitus (*n* = 77), Atrial Fibrillation (*n* = 20), Alcohol (*n* = 49), Smoking (*n* = 58) | Single Antiplatelet (*n* = 68), Dual Antiplatelets (*n* = 95), Anticoagulants (*n* = 5), Both Antiplatelets and Anticoagulants (*n* = 22) | 58 | 14 days | Fazekas score | Cranial CT or MRI within 72 h of symptom onset and again whenever HT was suspected or within 14 days after stroke onset | n/a | Atrial fibrillation, infarct diameter and WMH are independent risk factors for HT. |
| *Early neurological deterioration or improvement (within 1st week)* | | | | | | | | | | |
| Feng, (2014) [43] | Small subcortical infarcts (<1.5 cm in diameter) Longitudinal, 435 | 71.1 ± 8.89, 219M/216F | Hypertension (*n* = 356), Diabetes (*n* = 118), Dyslipidemia (*n* = 218), Coronary Artery Disease (*n* = 69), Smoking (*n* = 87) | n/a | n/a | 1 week | ARWMC scale | n/a | NIHSS on admission and every day for a week | Age, diabetes mellitus, HbA1c and TC levels, baseline NIHSS score and LA severity are independently associated with END after small subcortical infarction. |
| Jeong, (2015) [44] | Single Small Subcortical Infarction, Longitudinal, 587 | 65 ± 12, 347M/240F BMI 25.0 ± 3.7 in END and 24.0 ± 3.3 in no END group | Hypertension (*n* = 412), Diabetes mellitus (*n* = 193), Hyperlipidemia (*n* = 147), Smoking (*n* = 248), Relevant artery stenosis (0% to 50% narrowing) (*n* = 159), branch atheromatous lesions (*n* = 220) | n/a | 113 | 3 weeks | Fazekas score | Within 24 h of admission | NIHSS on admission and at least once a day during hospitalization, mRS at 3 months | END is associated with large vessel pathologies, but not previous lacunar infarction, CMBs or WMHs. |
| Nannoni, (2015) [45] | Subcortical stroke, Longitudinal, 94 | 67.5 ± 11.7, 68M/26F | Hypertension (*n* = 76), Diabetes mellitus (*n* = 22), Hypercholesterolaemia (*n* = 32), Coronary Artery Disease (*n* = 7), Smoking (*n* = 66) | n/a | 22 | 72 h | Van Swieten Scale and Fazekas score | Baseline CT scan on admission. Progressors had CT or MRI performed within 24 h from the onset of worsening. Non-progressors had CT or MRI performed before discharge | NIHSS on admission, every 6–8 h during hospitalization, at discharge. | Combined vascular risk factors, infarct size and severe LA are independently associated with END |

**Table 1.** *Cont.*

| 1st Author, (Year) | Type of Stroke, Study Design, Participants (*n*) | Demographics: Age (Years), Gender (M/F), Education (Years or Level), Marital/Occupational Status, Income, BMI | Cerebrovascular Risk Factors (*n*) | Medication (*n*) | Previous Stroke (*n*) | Follow up Time | Leukoaraiosis/ WMH Assessment | Time of MRI | Scale of Stroke Severity and Prognosis/Clinical Outcome | Main Findings |
|---|---|---|---|---|---|---|---|---|---|---|
| Nam, (2016) [46] | Isolated pontine infarction, Longitudinal, 82 | 71, 51M/31F | Hypertension (*n* = 59), Diabetes mellitus (*n* = 39), Hyperlipidemia (*n* = 39), Smoking (*n* = 37) | Mono antiplatelet drug (*n* = 48), Dual antiplatelet drug (*n* = 31), Anticoagulant (*n* = 0), Both (*n* = 2), None (*n* = 1) | 13 | 72 h | Fazekas score | Within 24 h from admission | NIHSS on admission and after 72 h from admission | Severe periventricular and subcortical WMHa are associated with END in patients with isolated pontine infarction. |
| Chen, (2017) [47] | AIS (with NIHSS < 12), Longitudinal, 687 | 59.6 ± 12.7, 496M/191F | Hypertension (*n* = 476), Diabetes mellitus (*n* = 162), Coronary Artery Disease (*n* = 34), Smoking (*n* = 243) | Mono-antiplatelet treatment (*n* = 391), Dual-antiplatelet treatment (*n* = 245), Anticoagulation treatment (*n* = 54), Statins (*n* = 48) | 111 | 72 h after admission | Fazekas score | During the initial 24 h after admission | NIHSS | END was not associated with SVD markers |
| Nam, (2017) [48] | AIS, Longitudinal, 325 | 69, 202M/123F | Hypertension (*n* = 232), Diabetes mellitus (*n* = 99), Hyperlipidemia (*n* = 100), Atrial Fibrillation (*n* = 89), Smoking (*n* = 104) | Mono-antiplatelet (*n* = 155) Dual-antiplatelet (*n* = 138) Anticoagulation (*n* = 10) Both 14 None 8 | | 4 days | Fazekas score | Within 24 h of admission | NIHSS on admission, at 72 h and 7 days, mRS at 3 months | END is associated with severe LA, distal hyperintense vessel sign, advanced age, history of atrial fibrillation and NIHSS on admission. |
| Marek, (2018) [49] | AIS, Longitudinal, 77 | 50–93, 44M/33F | n/a | n/a | n/a | n/a | van Swieten scale | Upon admission | NIHSS on admission | Extent of LA is not correlated with NIHSS scale upon admission. |
| Etherton, (2019) [50] | AIS, Longitudinal, 42 | 70.2 ± 9.2, 24M/18F | Atrial fibrillation (*n* = 16), Myocardial infarction (*n* = 7), Diabetes mellitus (*n* = 9), Hyperlipidemia (*n* = 23), Hypertension (*n* = 29), Smoking (*n* = 26) | | 7 | 5 days | WML volumetry | Within 12 h and at day 3–5 post stroke | NIHSS on admission and at day 3–5 post stroke | Periventricular WMHs and preserved normal appearing white-matter microstructural integrity are associated with early neurological improvement. |
| Nam, (2021) [51] | AIS (cryptogenic stroke patients with active cancer), Longitudinal, 179 | 67 ± 10, 108M/71F | Hypertension (*n* = 77), Diabetes mellitus (*n* = 55), Dyslipidemia (*n* = 76), Smoking (*n* = 61) | n/a | n/a | 3 months | Fazekas score | n/a | NIHSS at baseline and 72 h, mRS at 3 months | No association between END and CMBs or WMHs was observed. In contrast, SBI were associated with END |
| *Early Stroke Outcome (<1 month)* | | | | | | | | | | |
| Kang, (2013) [52] | AIS, Longitudinal, 408 | 64.7 ± 10.0, 238M/170F, 8.5 ± 5.0 | Hypertension (*n* = 204), Diabetes mellitus (*n* = 112), Heart disease (*n* = 33), Hypercholesterolemia (*n* = 229) | n/a | 34 | 2 weeks, 1 year for 284 patients | Fazekas score | At admission | NIHSS, mRS, BI on admission, at 2 weeks and at 1 year, MMSE at the 2 weeks and 1 year. | Severe periventricular WMHs, but not deep WMHs, are associated with worse outcome both at 2 weeks and at 1 year after stroke. |

**Table 1.** *Cont.*

| 1st Author, (Year) | Type of Stroke, Study Design, Participants (*n*) | Demographics: Age (Years), Gender (M/F), Education (Years or Level), Marital/Occupational Status, Income, BMI | Cerebrovascular Risk Factors (*n*) | Medication (*n*) | Previous Stroke (*n*) | Follow up Time | Leukoaraiosis/ WMH Assessment | Time of MRI | Scale of Stroke Severity and Prognosis/Clinical Outcome | Main Findings |
|---|---|---|---|---|---|---|---|---|---|---|
| Huang, (2014) [53] | Acute cerebral infarction, Longitudinal, 279 | 73.11 ± 10.32, 191M/88F | n/a | n/a | n/a | 14 days or until discharge | Fazekas score | Before or within 5 days after admission | NIHSS on days 1, 7 and 14 after admission and on the day of discharge, mRS on the day of discharge | WMH grade ≥2 is associated with high probability of self-care incapability upon discharge |
| Toscano, (2015) [54] | Any type (205 with ischemic stroke), Longitudinal, 254 | 73 ± 11.6, 138M/116F, years of education: 0–4 (*n* = 23), 5–7 (*n* = 115), 8–12 (*n* = 72), 13+ (*n* = 44) | Coronary Artery Disease (*n* = 217) | n/a | | 14 days | Fazekas score | Within 7 days from stroke onset | NIHSS on admission, Water swallow test upon admission and after 14 days | Age, NIHSS ≥ 12 upon admission and LA degree are associated with early post-stroke dysphagia |
| Zhang, (2017) [55] | AIS (with NIHSS score ≤ 5), Longitudinal, 217, 147 females | 70M/147F, None-to-mild leukoaraiosis group vs. moderate leukoaraiosis group: illiterate (30% vs. 31%), primary school (26.7% vs. 29%), middle school or higher (43.3% vs. 40%) | Hypertension (*n* = 156), Diabetes mellitus (*n* = 111), Dyslipidemia (*n* = 53), Coronary artery disease (*n* = 13), Smoking (*n* = 90) | Antiplatelets (*n* = 210), Oral anticoagulants (*n* = 4), Statins (*n* = 217), Antiglycemics (*n* = 53), Antihypertensives (*n* = 95) | n/a | 30 days | Fazekas score | Within 48 h after admission | NIHSS and MMSE at baseline and at 30 days. | Severe LA is associated with poor early functional recovery. |
| Fandler, (2017) [56] | Recent small subcortical infarct, Longitudinal, 332 | 67.7 ± 11.9, 214M/118F | Hypertension (*n* = 282), Diabetes mellitus (*n* = 93), Hyperlipidemia (*n* = 197), Smoking (*n* = 97) | n/a | 73 | 1 day (range, 0–9) | Fazekas score | n/a | NIHSS on admission Gugging Swallowing Screen | NIHSS on admission, pontine infarcts and severe baseline WMHs are independent risk factors of early post-stroke dysphagia |
| Ko, (2017) [57] | Acute Unilateral Corona Radiata Infarction with or without contralateral CBT Involvement, Longitudinal, 81, 29 females | 64.6 ± 11.5, 344M/115F | Hypertension (*n* = 43), Diabetes mellitus (*n* = 19) | n/a | n/a | Until discharge | Fazekas score | n/a | NIHSS at baseline, bedside swallowing test, videofluoroscopic swallowing study | Contralateral CBT involvement predicts post-stroke dysphagia. |
| Shang, (2020) [58] | AIS (patients with MCA stroke), Longitudinal, 459 | n/a | Hypertension (*n* = 245), Diabetes mellitus (*n* = 109), Hypercholesteremia (*n* = 121), Smoking (*n* = 149) | Antiplatelet drugs (*n* = 176) Anticoagulants (*n* = 8) Statins (*n* = 153) | 81 | At discharge | Fazekas score | Two groups: ≤14-day and ≥15-day | NIHSS on admission, mRS at discharge | LA combined with fluid attenuated inversion recovery vascular hyperintensity are associated with unfavorable early clinical outcomes |
| Li, (2020) [59] | Any type (98 with ischemic stroke), Longitudinal, 130 | 57.87 ± 14.22, 83M/47F, no education (*n* = 5), some education (*n* = 124), never married (*n* = 7), currently married (*n* = 111), formerly married (*n* = 12), high income (*n* = 30), low income (*n* = 100) | Hypertension (*n* = 120), Diabetes mellitus (*n* = 45), Dyslipidemia (*n* = 109), Alcohol (*n* = 54) | n/a | n/a | 12 months (only the one-month endpoint was used for analysis) | WML volumetry | Within 10 days of symptom onset | mRS, NIHSS, MBI | WMH volume is positively correlated to poor 1-month post-stroke outcome |

**Table 1.** *Cont.*

| 1st Author, (Year) | Type of Stroke, Study Design, Participants (*n*) | Demographics: Age (Years), Gender (M/F), Education (Years or Level), Marital/Occupational Status, Income, BMI | Cerebrovascular Risk Factors (*n*) | Medication (*n*) | Previous Stroke (*n*) | Follow up Time | Leukoaraiosis/ WMH Assessment | Time of MRI | Scale of Stroke Severity and Prognosis/Clinical Outcome | Main Findings |
|---|---|---|---|---|---|---|---|---|---|---|
| Kim, (2014) [60] | AIS, Longitudinal, 225 | 67.6 ± 13.7, 123M/102F | Hypertension (*n* = 141), Diabetes mellitus (*n* = 64), Dyslipidemia (*n* = 87), Ischemic Heart Disease (*n* = 19), Atrial Fibrillation (*n* = 53) | Anticoagulants (*n* = 4), Antiplatelets(*n* = 48), Statins (*n* = 12) | 44 | 6 months | Semiquantitative Scheltens visual rating system [61] | Within 24 h of admission | NIHSS on admission, mRS at discharge and at 6 months | Poor functional outcome upon discharge and at 6 months is positively associated with advanced age, presence of CMB and WMH scores. |
| *Short-term stroke outcome (1–3 months)* | | | | | | | | | | |
| Onteddu, (2015) [62] | AIS (with NIHSS ≤ 5), Longitudinal, 185 | 69, 119M/66F | Hypertension (*n* = 138), Dyslipidemia (*n* = 110), Diabetes (*n* = 63), Atrial fibrillation (*n* = 36), Coronary artery disease (*n* = 51), Congestive heart failure (*n* = 21), Peripheral vascular disease (*n* = 13), Smoking (*n* = 56) | Antiplatelets (*n* = 100), Oral anticoagulants (*n* = 15), Statins (*n* = 85), Antiglycemics (*n* = 46), Antihypertensives (*n* = 125) | 37 | 90 days | van Swieten scale, Fazekas score | n/a | NIHSS at baseline, discharge, and 90 days, mRS at 90 days | Compared to chronological age, LA is a more sensitive predictor of short term neurological deficit after minor ischemic stroke. |
| Helenius, (2017) [63] | Acute small subcortical infarct, Longitudinal, 80 | 67, 43M/37F | Hypertension (*n* = 57), Dyslipidemia (*n* = 49), Diabetes mellitus (*n* = 27), Atrial fibrillation (*n* = 5), Coronary Artery Disease (*n* = 18), Peripheral artery disease (*n* = 8), Smoking (*n* = 19), Alcohol (*n* = 9) | Statins (*n* = 42), Antihypertensives (*n* = 52), Antiglycemics (*n* = 23), Antiplatelets (*n* = 48), Oral anticoagulant (*n* = 4) | 20 | 90 days | Fazekas score | Between 24 and 168 h since symptom onset | NIHSS at presentation, mRS at presentation and at 90 days. | Greater WMH burden was independently associated with SSI infarct volume and a worse 90-day functional outcome. |
| Ryu, (2017) [64] | AIS (large artery atherosclerosis, small vessel occlusion, or cardioembolism) Longitudinal, 5035 | 66.3 ± 12.8, 3000M/2032F, BMI per WMH quantiles: Q1 = 23.9, Q2 = 23.9, Q3 = 23.8,Q4 = 23.4, Q5 = 23.3 | Hypertension (*n* = 3250), Hyperlipidemia (*n* = 1587), Diabetes mellitus (*n* = 1363), Smoking (*n* = 1951), Coronary Artery Disease (*n* = 425), Atrial Fibrillation (*n* = 989) | Statins (*n* = 558), Antiplatelets (*n* = 1049) | n/a | 3 months | WML volumetry | n/a | NIHSS on admission, mRS pre-stroke, at discharge and at 3 months. | Higher WMH volumes are associated with poor short-term prognosis mainly in large artery atherosclerosis stroke |

**Table 1.** *Cont.*

| 1st Author, (Year) | Type of Stroke, Study Design, Participants (*n*) | Demographics: Age (Years), Gender (M/F), Education (Years or Level), Marital/Occupational Status, Income, BMI | Cerebrovascular Risk Factors (*n*) | Medication (*n*) | Previous Stroke (*n*) | Follow up Time | Leukoaraiosis/ WMH Assessment | Time of MRI | Scale of Stroke Severity and Prognosis/Clinical Outcome | Main Findings |
|---|---|---|---|---|---|---|---|---|---|---|
| Giralt-Steinhauer, (2018) [65] | TIA or AIS, Longitudinal, 313 | 200M/113F | Hypertension (*n* = 232), Hyperlipidemia (*n* = 180), Diabetes mellitus (*n* = 124), Coronary Artery Disease (*n* = 37), Atrial Fibrillation (*n* = 76), Peripheral artery disease (*n* = 36) | n/a | 27 | 3 months | WML volumetry | Median 7 days | NIHSS on admission, mRS at 3 months | Brainstem WMHs are independent predictors of poor outcome after AIS/TIA. |
| Jeong, (2018) [66] | AIS (with cryptogenic stroke), Longitudinal, 235 | 63 (IQR, 54–69), 159M/76F | Hypertension (*n* = 178), Hyperlipidemia (*n* = 13), Diabetes mellitus (*n* = 72), Smoking (*n* = 118) Cerebral artery atherosclerosis: intracranial (*n* = 61), extracranial (*n* = 23), combined (*n* = 46) | n/a | n/a | A median of 7.7 years (6.7–9.0) | Fazekas score | Within 3 days after admission | NIHSS at baseline, mRS at 3 months | Severe WMHs are associated with poor short-term functional outcome in young and old CS patients. Younger CS patients with severe WMHs had higher death rates |
| Zerna, (2018) [67] | High risk TIA or AIS (NHISS ≤ 3), Longitudinal, 412 | 69, 248M/164F | Hypertension (*n* = 225), Congestive heart failure (*n* = 3), Atrial fibrillation (*n* = 28), Diabetes mellitus (*n* = 57), Smoking (*n* = 63), Intracranial occlusion/stenosis (*n* = 64) | Aspirin (*n* = 134), Plavix (*n* = 24), Aggrenox (*n* = 4), Warfarin (*n* = 17) | n/a | 90 days | Fazekas score, WML volumetry | n/a | NIHSS on admission, mRS at 90 days | WML burden was associated with short-term outcomes in TIA and minor stroke patients who had good prestroke function in the presence of intracranial stenosis/occlusion. |
| Schirmer, (2018) [68] | AIS, Longitudinal, 453 | 66.6 ± 14.7, 165M/288F | Hypertension (*n* = 311) | n/a | n/a | 3–6 months | WML volumetry | Within 48 h of admission | mRS at 90 days | Significant direct association between WMH volume and early stroke outcome was not established. |
| Appleton, (2020) [69] | Any type (3342 with ischemic stroke), Longitudinal, 4011 | 70.3, 2297M/1714F | Hypertension (*n* = 2607), Diabetes mellitus (*n* = 699), Atrial Fibrillation (*n* = 762), Ischemic Heart Disease (*n* = 669), Peripheral Arterial Disease (*n* = 117), Smoking (*n* = 945), Hyperlipidemia (*n* = 1098), Alcohol (*n* = 294) | n/a | 1138 | 90 days | LA was assessed separately in anterior and posterior brain regions defined as 0 = no lucency, 1 = lucency restricted to region adjoining ventricles, or 2 = lucency covering entire region from lateral ventricle to cortex | At baseline, usually before randomization | NIHSS at baseline, mRS at day 90 | Severe LA, severe cerebral atrophy and old lacunar infarcts/lacunes are associated with unfavorable functional outcome at day 90 in lacunar and non-lacunar stroke, with a stronger effect in lacunar stroke. |

**Table 1.** *Cont.*

| 1st Author, (Year) | Type of Stroke, Study Design, Participants (*n*) | Demographics: Age (Years), Gender (M/F), Education (Years or Level), Marital/Occupational Status, Income, BMI | Cerebrovascular Risk Factors (*n*) | Medication (*n*) | Previous Stroke (*n*) | Follow up Time | Leukoaraiosis/ WMH Assessment | Time of MRI | Scale of Stroke Severity and Prognosis/Clinical Outcome | Main Findings |
|---|---|---|---|---|---|---|---|---|---|---|
| Griessenauer, (2020) [70] | AIS, Large vessel occlusion (LVO) and non-LVO Longitudinal, 1285 | 69 (58–78), 497M/788F BMI ≥ 30 (n = 755) | Hypertension (*n* = 1264), Diabetes mellitus (*n* = 621), Dyslipidemia (*n* = 1148), Smoking (*n* = 954), Alcohol (*n* = 552), Peripheral vascular disease (*n* = 161), Coronary artery disease (*n* = 477), Atrial fibrillation (*n* = 355), Carotid stenosis (*n* = 607), Intracranial atherosclerotic stenosis (*n* = 483), Anemia (*n* = 243), Sleep apnea (*n* = 170), COPD (*n* = 188) | n/a | 503 | 90 days | WML volumetry | At the time of the stroke admission | NIHSS at presentation, mRS at 90 days | Increasing WMH volume from 0 to 4 mL is correlated with an unfavorable outcome among LVO and non-LVO patients |
| Ryu, (2020) [71] | AIS, Longitudinal, 477 | 66 ± 14, 294M/183F | Hypertension (*n* = 359), Diabetes mellitus (*n* = 182), Hyperlipidemia (*n* = 143), Smoking (*n* = 221), Coronary Artery Disease (*n* = 75) | n/a | 99 | 3 months | WML volumetry | Within 7 days of stroke onset | NIHSS on admission, mRS at 3 months | WMHs, lacunes and CMBs are independently associated with mRS scores at 3 months. |
| Coutureau, (2021) [72] | AIS, Longitudinal, • Dataset 1: 348, Dataset 2: 137 | Dataset 1: 67.5 ± 14.1, 221M/127F, Baccalaureate or higher educational status (*n* = 106) Dataset 2: 64.8 ± 12.6, 82M/55F, Baccalaureate or higher educational status (*n* = 51) | | n/a | n/a | 3 months for dataset 1 and 6 months for dataset 2 | Fazekas score | 24 and 72 h after stroke onset | NIHSS at baseline, mRS at 3 months for dataset 1 and mRS at 6 months for dataset 2 | Total SVD score was associated with poor early recovery post-stroke, but did not provide significant improvement of prediction models compared to age and baseline NIHSS |
| Farag, (2021) [73] | AIS, Longitudinal, 460 | 282M/178F | Hypertension (*n* = 319), Diabetes mellitus (*n* = 243), Smoking (*n* = 186), Ischemic Heart Disease (*n* = 109) | Antiplatelets (*n* = 243), Statins (*n* = 54) | 174 | 3 months | Group A (absent leukoaraiosis) and group B (present leukoaraiosis). | A few days post-admission | NIHSS at baseline, mRS at discharge and at 3 months | LA degree is not associated with stroke severity on admission, but with worse clinical outcome at 3 months follow up. |
| Bu, (2021) [61] | AIS, Longitudinal, 259 | 69 ± 12, 139M/120F | n/a | n/a | 42 | 90 days | Fazekas score | Within 9 h of symptom onset | NIHSS at baseline, mRS at 90 days | WMHs were not associated with short-term stroke outcome. |

**Table 1.** *Cont.*

| 1st Author, (Year) | Type of Stroke, Study Design, Participants (*n*) | Demographics: Age (Years), Gender (M/F), Education (Years or Level), Marital/Occupational Status, Income, BMI | Cerebrovascular Risk Factors (*n*) | Medication (*n*) | Previous Stroke (*n*) | Follow up Time | Leukoaraiosis/ WMH Assessment | Time of MRI | Scale of Stroke Severity and Prognosis/Clinical Outcome | Main Findings |
|---|---|---|---|---|---|---|---|---|---|---|
| Sakuta, (2021) [74] | Non-cardiogenic AIS (NIHSS score < 4), Longitudinal, 240 | 66 (57–76), 187M/53F BMI (median): 23.7 | Hypertension (*n* = 165), Diabetes mellitus (*n* = 74), Dyslipidemia (*n* = 126), Ischemic Heart Disease (*n* = 20), Peripheral arterial disease (*n* = 6), Chronic kidney disease (*n* = 33), Malignant neoplasms (*n* = 17) | Single antiplatelet agent (*n* = 150), Dual antiplatelet agent (*n* = 90) | 43 | 90 days | Fazekas score | On admission | NIHSS on admission, mRS at day 90 | deep WMH is not an independent risk factor for poor short-term functional outcome |
| Chen, (2021) [75] | AIS (with NIHSS ≤ 3), Longitudinal, 388 | 66.54 ± 11.15, 111M/277F | Hypertension (*n* = 324), Coronary artery disease (*n* = 51), Atrial fibrillation (*n* = 52), Diabetes mellitus (*n* = 142), Hyperlipidemia (*n* = 162), Stroke (*n* = 61), Smoking 73, Alcohol (*n* = 56), Carotid atherosclerosis (*n* = 259) | n/a | n/a | 90 days | Fazekas score | Within 7 days of stroke onset | NIHSS on admission, mRS at 90 days | Among SVD neuroimaging markers, only Fazekas score was associated with poor short-term outcome in minor ischemic stroke. |
| Song, (2021) [76] | TIA, TSI or AIS with symptomatic carotid artery stenosis (CAS), Longitudinal, 158 | 134M/24F, BMI: TIA/TSI = 23.6 3.9, Mild = 23.8 ± 4.0, Moderate to severe = 23.4 ± 2.7 | Hypertension (*n* = 123), Diabetes mellitus (*n* = 59), Hyperlipidemia (*n* = 78), Ischemic Heart Disease (*n* = 37), Peripheral arterial disease (*n* = 13), Smoking (*n* = 53), Congestive Heart Failure (*n* = 22), Atrial Fibrillation (*n* = 10) | Antiplatelets (*n* = 93), Anticoagulants (*n* = 10), Statins (*n* = 82) | 45 | 90 days | WML volumetry and Fazekas scale | Within 48 h from the symptom onset | NIHSS at baseline, mRS at day 90 | Larger WMH volume, but not brain-blood flow dynamics or carotid plaque characteristics, is associated with moderate to severe stroke and poor short-term prognosis in symptomatic CAS patients. |
| *Long-term stroke outcome (3–12 months)* | | | | | | | | | | |
| Reid, (2012) [77] | Any type (468 with ischemic stroke), Longitudinal, 538 | 74 (61–80), 286M/252F | Atrial Fibrillation (*n* = 78) | n/a | 148 | 6 months | Score previously described by Reid et al. [74] | n/a | mRS | LA score was the only independent radiological predictor of both excellent and devastating outcomes. |

**Table 1.** *Cont.*

| 1st Author, (Year) | Type of Stroke, Study Design, Participants (*n*) | Demographics: Age (Years), Gender (M/F), Education (Years or Level), Marital/Occupational Status, Income, BMI | Cerebrovascular Risk Factors (*n*) | Medication (*n*) | Previous Stroke (*n*) | Follow up Time | Leukoaraiosis/ WMH Assessment | Time of MRI | Scale of Stroke Severity and Prognosis/Clinical Outcome | Main Findings |
|---|---|---|---|---|---|---|---|---|---|---|
| Liu, (2017) [78] | AIS; patients with branch atheromatous disease (BAD), Longitudinal, 176 | 121M/55F | Hypertension (*n* = 132), Diabetes mellitus (*n* = 63), Dyslipidemia (*n* = 101), Ischemic Heart Disease (*n* = 14), Smoking (*n* = 78), Alcohol (*n* = 67) | n/a | n/a | 6 months | Fazekas score | Within 5 days of admission | NIHSS, mRS at 6 months | WMHs were associated with 6-month functional outcome only in the paramedian pontine and not the lenticulostriate atherosclerotic cerebral infarction group |
| Wardlaw, (2017) [79] | AIS (with NIHSS ≤ 7), Longitudinal, 190 | 65.3 ± 11.3, 112M/78F | Hypertension (*n* = 142), Smoking (*n* = 73), Hyperlipidemia (*n* = 116), Diabetes mellitus (*n* = 21) | n/a | n/a | 1 year | Fazekas score | At presentation and at 1 year after stroke | NIHSS on admission, mRS at 1 year | WMH may regress after minor stroke. Blood pressure reduction might accentuate WMH growth |
| Auriat, (2019) [39] | Any type, Cross-sectional, 30 | 65.68 ± 8.16, 21M/7F, 14.47 ± 3.23 | n/a | n/a | n/a | Last diagnosed stroke at least 6 months before participation in the study | WML volumetry | n/a | The upper-extremity motor portion of the Fugl-Meyer assessment (FM) to index impairment of the hemiparetic arm and Wolf Motor Function Test to assess motor function of the upper extremity. | deep WMH volume correlated with motor function and impairment. Periventricular WMH volume is associated with non-memory impairment |
| Hicks, (2018) [80] | Any type (with upper extremity hemiparesis), Longitudinal, 28 | 63.2 ± 11.5 | n/a | n/a | n/a | post-stroke interval more than 10 months | WML volumetry | Pre-treatment | Hemiparetic arm function was measured using the Motor Activity Log and Wolf Motor Function Test | WMH predicts 10-month stroke-related upper extremity motor impairment. |
| Wright, (2018) [40] | AIS, Cross-sectional, 42 | 56.1 ± 15.0 | n/a | n/a | n/a | >3 months | CHS rating scale | 1 to 4 weeks from stroke onset | WAB object naming score and WAB word fluency score | >3-months post-stroke naming outcome is associated with LA severity |
| Lee, (2020) [81] | AIS, Longitudinal, 137 | 68.7 ± 14.0 69M/68F | Diabetes mellitus (26.2%), Hypertension (71.4%), Smoking (9.5%). | n/a | n/a | 6 months | Fazekas score | n/a | NIHSS on admission, Clinical Dysphagia Scale at baseline | WMH burden is associated with 6-month post-stroke dysphagia |
| *Chronic stroke outcome (>1 year)* | | | | | | | | | | |
| Baik, (2017) [82] | AIS (with large artery atherosclerosis), Longitudinal, 538 | 65.7 ± 10.3, 305M/233F | Hypertension (*n* = 416), Hyperlipidemia (*n* = 68), Diabetes mellitus (*n* = 221), Smoking (*n* = 268) | Antithrombotics (*n* = 40), Statins (*n* = 24) | n/a | Median of 7.7 years (interquartile range, 5.6–9.7) | Fazekas score | n/a | NIHSS at baseline, mRS at 3 months | WMH severity is associated with increased all-cause and cardiovascular mortality. |

**Table 1.** *Cont.*

| 1st Author, (Year) | Type of Stroke, Study Design, Participants (*n*) | Demographics: Age (Years), Gender (M/F), Education (Years or Level), Marital/Occupational Status, Income, BMI | Cerebrovascular Risk Factors (*n*) | Medication (*n*) | Previous Stroke (*n*) | Follow up Time | Leukoaraiosis/ WMH Assessment | Time of MRI | Scale of Stroke Severity and Prognosis/Clinical Outcome | Main Findings |
|---|---|---|---|---|---|---|---|---|---|---|
| Jeon, (2017) [83] | AIS, Longitudinal, 1138 | 73.3 ± 9.9, 156M/982F | Hypertension (*n* = 192), Diabetes mellitus (*n* = 90), Coronary heart disease (*n* = 3), Dyslipidemia (*n* = 79), Smoking (*n* = 49) | n/a | 63 | 3 years | Fazekas score | At the time of admission | 3-year mortality | SVD, especially, WMH, and renal dysfunction are associated with increased 3-year mortality post-stroke |
| Hert, (2020) [84] | Atrial fibrillation stroke (treated with anticoagulation), Longitudinal, 320 | 78.2 ± 9.2, 170M/150F | Hypertension (*n* = 241), Diabetes mellitus (*n* = 62), Hypercholesterolemia (*n* = 122), Smoking (*n* = 81), Alcohol (*n* = 78) | Vitamin K Antagonists (*n* = 61), Vitamin K Antagonists/Antiplatelet (*n* = 15), Direct Oral Anticoagulants (*n* = 216), Direct Oral Anticoagulants/antiplatelet (*n* = 18), Antiplatelet (*n* = 5) | n/a | Median follow-up time of 754 days | ARWMC score | n/a | NIHSS at baseline, mRS at 3, 6, 12, and 24 months | WMHs and CMBs were related to increased risk of ischemic stroke, intracranial hemorrhage, death and disability 2 years post stroke. |
| *Rehabilitation outcome* | | | | | | | | | | |
| Senda, (2016) [85] | AIS, Longitudinal, 520 | 72.8 ± 8.4, 317M/203F | Hypertension (*n* = 325), Diabetes mellitus (*n* = 147), Hyperlipidemia (*n* = 183), Smoking (*n* = 204), History of Heart Disease (*n* = 105) | n/a | 131 | From admission to discharge from a convalescent rehabilitation hospital | Fazekas score | n/a | FIM scores on admission and at discharge | Periventricular WMHs are associated with poor FIM motor scores, whereas deep WMHs are related to poor FIM cognitive scores at discharge from rehabilitation. |
| Khan, (2019) [86] | AIS, Longitudinal, 109 | 66.6 ± 12.4, 65M/44F | Hypertension (*n* = 84), Diabetes mellitus (*n* = 43), Atrial Fibrillation (*n* = 22) | n/a | n/a | From admission to discharge from acute inpatient rehabilitation | Fazekas score | n/a | NIHSS at baseline, Functional Independence Measure (FIM) motor and cognitive score | LA severity is an independent predictor of cognitive, but not motor FIM score, after rehabilitation for AIS. |
| Dai, (2022) [87] | Any type (172 with ischemic stroke), Longitudinal, 210 | 67.3, 142M/68F, BMI: no/mild WHMs = 24.5, moderate/severe WMHs = 26.2 | Hypertension (*n* = 128), Diabetes mellitus (*n* = 43), Dyslipidemia (*n* = 98), Smoking (*n* = 121), Alcohol (*n* = 49) | n/a | n/a | Until discharge from the neurorehabilitation ward | Fazekas score | NM | NIHSS on admission, mRS, Postural Assessment Scale for Stroke and modified Fugl–Meyer Gait Assessment on day 30 ± 3 post-stroke and at discharge from the rehabilitation ward | LA severity is independently associated with poor gait and balance recovery and increased risk of falls post-AIS. |

### 3.6. Stroke Patient Groups and Demographic Profile

The total number of stroke patients included in all studies ranges from *n* = 28 [18,80] to *n* = 5035 [19,64]. Across the 49 studies, 9 studies have a disease sample size between 1–100 patients, 9 studies between 101–200, 9 studies between 201–300, 5 studies between 301–400, 8 studies between 401–500 and 9 studies have a disease sample size larger than 500 patients. Mean/median patients' age ranges from 56.1 years [17,40] to 78.2 years [20,84].

### 3.7. Reference Groups

In none of the 49 included studies, stroke patients are contrasted to demographically—matched healthy individuals and none of the studies include a disease-control group other than stroke patients.

### 3.8. Time of MRI Execution

In 5 studies the MRI scanning was performed upon admission, in 1 study in the first 9 h, in 1 study in the first 12 h, in 7 studies in the first 24 h, in 3 studies within the first 48 h, in 2 studies in the first 72 h, in 3 studies the MRI was executed within 5 days from admission, in 7 studies within the first week, in 1 study in the first 10 days and lastly, in 1 study the MRI was performed in the first month.

### 3.9. Scales of Stroke Severity and Prognosis/Clinical Outcome

NIHSS and mRS were simultaneously used in 25 studies. NIHSS was the only scale in 8 studies and mRS exclusively in 2 studies. In 1 study the FIM score was exclusively utilized, in 1 study the Wolf Motor Function Test and in 1 study the WAB score. In the rest of the studies there was a combination of scales of stroke severity and clinical outcome. More specifically, in 1 study NIHSS was combined with GCS, in 1 study with MMSE, in 1 study with the Water Swallow Test, in 1 study with the Clinical Dysphagia Scale, in 1 study with the FIM score, in 1 study with mRS and mBI, in 1 study with mRS, BI and MMSE and in 1 study with mRS, Postural Assessment Scale and modified Fugl-Meyer Gait assessment. Lastly, 1 study combined the upper-extremity motor portion of the Fugl-Meyer assessment to index impairment of the hemiparetic arm and Wolf Motor Function Test to assess motor function of the upper extremity.

## 4. Discussion

A literature review over the last decade was conducted to elucidate the prognostic value of LA after AIS. Forty-nine full-text original articles dealing with the potential utility of the assessment of pre-existing leukoaraiosis in AIS patients' prognosis were identified and divided into seven groups.

### 4.1. Hemorrhagic Transformation

Even though an association between LA and hemorrhagic transformation (HT) risk after AIS has been proposed for decade, the underlying mechanistic link remains unclear. Moderate to severe LA indicates vessel fragility and chronic injury resulting from BBB alterations due to chronic brain edema and genetic factors. HT is also believed to result from abnormal BBB permeability, which allows extravasation of blood products in or out of the ischemic region after restoration of blood flow. Thus, by sharing common underlying pathophysiological mechanisms linked to BBB disruption, preexisting LA may facilitate HT pathogenesis [24].

Better understanding of which patients with cardioembolic stroke are at higher risk of hemorrhagic transformation (HT) after AIS is crucial for maximizing the safety and efficacy of oral anticoagulants utilized for stroke prevention, but also guide more rational use of thrombolysis which may not be used for fear of HT. Wei and colleagues [41] studied patients with cardioembolic stroke of atrial fibrillation or rheumatic heart disease origin and revealed that HT risk was not related to age, gender, history of hypertension or blood pressure on admission, DM, hyperlipidemia, prior stroke, smoking or alcohol consumption

or therapy with antiplatelets, anticoagulants or statins. In contrast, HT risk was associated with higher median NIHSS score and lower median Glasgow Coma Scale (GCS) score, larger infarct areas, lower levels of serum total cholesterol or low-density lipoprotein on admission. Moreover, researchers exhibited that moderate to severe LA at frontal or periventricular sites, but not parieto-occipital, temporal or basal ganglia, increased HT risk by three times, thus suggesting that LA at those sites should be considered before proceeding to anticoagulation treatment or thrombolysis after AIS.

Compared to anterior circulation infarctions, HT complicating infarctions involving the cerebellum are more likely to induce neurological deterioration due to brainstem repression affecting vital signs and fourth ventricle compression resulting in obstructive hydrocephalus. Therefore, investigating factors correlated with HT in acute cerebellar infarction is critical. Wang and colleagues [42] recruited 190 patients with cerebellar infarction of whom 37 exhibited HT within the next 14 days. Researchers did not identify age, gender, hypertension, DM, smoking, drinking, previous stroke, levels of fasting blood glycose, platelet (PLT), monocyte count, total cholesterol (TC), low-density lipoprotein cholesterol (LDL-C) and high-density lipoprotein cholesterol (HDL-C), infarct site, enlarged perivascular spaces (EPVSs) as HT risk factors. In contrast, neutrophil count was significantly higher in the HT group and in keeping of their role as important mediators of acute ischemic brain injury via pro-inflammatory cytokine production and induction of MMP-9 expression leading to BBB destruction, infarct growth and HT. Furthermore, lymphocyte count was lower in the non-HT group solidifying T-lymphocyte role as brain's main protective immune modulators that eliminate inflammatory response [42]. Finally, independent association between higher grade WMHs and HT risk was shown. The authors proposed that by recognizing risk factors associated with cerebellar HT timely, clinicians can take effective measures to minimize the incidence of HT, thus improve AIS prognosis.

### 4.2. Early Neurological Deterioration

LA and small subcortical infarcts (SSIs) share similar pathogenetic mechanisms, as both are related to hypertensive hyaline degeneration, demyelination, and amyloid angiopathy. Moreover, LA being a marker of chronic ischemia resulted by compromised tissue perfusion, nerve fiber loss and reduced neuronal connectivity and chronic toxic edema due to BBB disruption, makes brain tissue more susceptible to damage and related early neurological deterioration (END) after SSIs. For patients with single small subcortical infarcts (<1.5 cm in diameter), traditionally called lacunar infarctions or small vessel occlusions, located in deep regions including the thalamus, gangliocapsular regions, corona radiata, or pons, Feng and colleagues [43] illustrated that age, DM, HbA1c and total cholesterol (TC) levels, baseline NIHSS score, total, global, and regional scores of LA severity were independently associated with END, defined as worsening at least by 2 points in the NIHSS score, or at least by 1 point in the NIHSS score for motor function within 1 week. Thus, the authors exhibited the importance of grading LA in clinical course evaluation and suggested that for patients with advanced LA more radical therapy with some brain-protecting agents and early rehabilitation might be more appropriate.

Similarly, Nannoni and colleagues [45] exhibited that in the case of minor subcortical strokes (>1.5 cm in diameter), END, defined as an increase of NIHSS motor score $\geq 1$ point within 72 h from onset, was associated with a high combined vascular risk profile (including hypertension, DM, smoking and hypercholesterolaemia), larger infarct size and severe LA. The pathophysiological link between severe LA and END has already been mentioned. With regard to infarct size, it is postulated that in contrast to small subcortical infarcts, larger infarcts could be due to branch atheromatous disease (BAD), associated with parent artery atheroma. Early neurological progression appears to occur more often among patients with BAD. The authors conclude that in case of minor subcortical infarcts early recognition of patients at risk of progression might be of importance to prevent clinical worsening. Defining END on a similar manner, Nam and colleagues [46] exhibited that severe periventricular and subcortical WMHs were associated with END in case of isolated

pontine infarction, as well. In contrast, END was not correlated with other demographic, clinical or laboratory factors. The authors suggest that entire brain cerebral perfusion decreases in patients with WMH, including areas of radiological normal appearance. Thus, propose that the pontine area may also share a hypoperfused state and the poor collateral perfusion may cause symptom progression in patients with severe periventricular and subcortical WMHs. The same group studied 325 AIS patients with symptomatic severe steno-occlusion of the middle cerebral or internal carotid artery that have not undergone thrombolytic therapy and revealed that END was positively associated with advanced age, higher initial NIHSS score, larger initial infarct volume, clinical history of atrial fibrillation, distal hyperintense vessel sign (HVS) and severe LA [48].

Etherton and colleagues [50] evaluated early neurological improvement (ENI), defined as a reduction in the NIHSS of ≥2 points from admission to day 3–5 post-stroke assessment stroke or a NIHSS score of 0 on follow-up assessment, among 42 consecutive AIS stroke patients. The authors revealed that ENI was associated with decreased rates of DM, hyperlipidemia or hypertension. Moreover, ENI was correlated to lower burden of periventricular WMH, but not deep WMH. Visible WMH represents radiographic injury to only a small fraction of total brain white matter. Thus, researchers checked normal appearing white matter (NAWM) structural integrity by applying NAWM diffusivity anisotropy metrics and showed that preserved NAWM structural integrity was also associated with increased odds of ENI.

On the contrary, Marek and colleagues [49] having analyzed 77 head CT scans of patients admitted from the emergency room to the Radiology Department due to suspected AIS, demonstrated that baseline LA severity was not correlated to NIHSS scale upon admission. Similarly, Chen and colleagues [47] evaluated 687 AIS patients with NIHSS score < 12 upon admission and failed to reveal a relationship between END and several imaging markers of small vessel disease (SVD), including silent lacunar infarction, deep CMBs, brain atrophy, basal ganglia and semiovale EPVSs and periventricular or semiovale WMHs. Furthermore, Jeong and colleagues [44] studied 587 AIS exhibiting single small subcortical infarctions and revealed that despite END association with large vessel pathologies, no correlation was evident between END and markers of SVD, such as previous lacunar infarction, CMBs or WMHs. Finally, Nam and colleagues [51] aiming to determine whether SVD imaging parameters are associated with the prognosis of cryptogenic stroke (CS) patients with active cancer, revealed that although END increased proportionally with an increase in the number of silent brain infarct (SBI) lesions, this correlation did not appear for CMBs or WMHs. The authors propose that WMHs, SBI, and CMBs encompass diverse pathophysiologies. Stroke due to thromboembolism occurs frequently in patients with active cancer. SBI being a surrogate marker for a high-risk group prone to thromboembolism may explain why only SBIs were associated with END in AIS patients that also suffer from active cancer.

### 4.3. Early Stroke Outcome (<1 Month)

With regard to early clinical outcome post AIS, Huang and colleagues [53] utilizing Fazekas score revealed that stroke patients with severe baseline LA (WMH grade ≥ 2) had a high probability of self-care incapability upon discharge. Moreover, NIHSS score improvement 7 days after admission and upon discharge was negatively associated with LA severity. Thus, the authors concluded that severe LA is associated with poor prognosis in elderly stroke patients and proposed that this could be attributed to the fact that patients with severe LA often also exhibit cerebrovascular disease and cognitive dysfunction, which may affect functional recovery. Additionally, severe LA has been linked to high incidence of pneumonia or urinary tract infections, which might also be related to poor prognosis.

Furthermore, Kim and colleagues [60] revealed that age, initial NIHSS and mRS on admission, presence of CMBs, especially lobar and infratentorial, and WMHs, as rated by using the semiquantitative visual rating system described by Scheltens and colleagues, were positively correlated with poor functional outcome upon discharge (1 month) and

6 months after admission. In contrast, there was no significant association between gender, preexisting DM, hypertension, hyperlipidemia, previous stroke, ischemic heart disease, stroke location and early or late functional outcome. With respect to poor early neurological outcome, the authors proposed that severe white matter changes probably reflect chronically reduced tissue perfusion and cerebrovascular reactivity, which may lead to reduced penumbra survival after AIS. With regard to poor late neurological outcome they suggested that patients with preexisting brain damage have less ability to compensate functionally than those without preexisting damage. Moreover, Li and colleagues [59] studied 130 West African patients and revealed that age, waist-to-hip ratio and platelet count were positively linearly related to WMH volume, whereas high WMH volume showed a strong association with poor early (1 month) stroke functional outcome. With respect to platelet count the authors propose that by playing a key role in the inflammatory pathway in the endothelium, platelets may be implicated in small vessel disease, as measured by WMH volumes.

Thrombolysis may not be provided to patients with mild neurological deficit. To identify which patients with first mild ischemic stroke (defined as baseline NIHSS score $\leq 5$) may experience substantial early functional disability, Zhang and colleagues [55] studied the link between baseline LA and NIHSS score on day 30 post-stroke. The authors revealed that LA burden is associated with worse early neurological outcome (higher NIHSS scores). Interestingly, increasing LA severity was associated with greater NIHSS deficits for similar sized infarcts. This could be explained by the fact that preexisting white matter impairment weakens brain plasticity and compensatory mechanisms after stroke. Moreover, it was revealed that short term functional recovery was not associated with infarct location. This could be attributed to preexisting connectivity interruption due to severe LA as even though structural damage from stroke may be local, remote dysfunction can occur in regions connected to the area of the lesion. Researchers propose that thrombolysis should be provided to such patients that present extensive LA, as these are more susceptible to poor functional recovery.

Kang and colleagues [52] evaluated whether periventricular WMHs or deep WMHs were associated with worse functional outcomes both at 2 weeks and at 1 year after stroke and concluded that the former, but not the latter, predicted poorer functional outcomes after stroke both in the acute and chronic phases. WMHs are related to decreased neuronal connectivity due to demyelination, loss of axons, and oligodendrocytes and subsequent decreased functional connectivity of distant cortical regions which could impair plasticity and inhibit recovery post-stroke. The authors hypothesize that deep WMHs primarily disrupt short association fibers while periventricular WMHs affect long association fibers, which mainly influence different degrees of plasticity in neural repair processes after stroke.

Apart from that, Shang and colleagues [58] revealed that LA severity combined with FLAIR vascular hyperintensity (FVH), both markers of poor collateral flow, in patients with middle cerebral artery (MCA) strokes predict poor early stroke outcome. LA is considered a composite marker of tissue perfusion and tissue susceptibility to ischemia, as research has exhibited links between LA severity and altered reactivity in response to reduced blood flow of penetrating, small arteries originating in pial vessels and supplying the white matter. Considering that FHV is likely related to stationary or slow blood, blockage of the upstream leptomeningeal collateral flow by narrow penetrating arteries could result in FVH and insufficient effective collateral flow, further inducing larger infarct volumes and unfavorable outcomes in patients with severe MCA stenosis or occlusion.

Early detection and management of post-stroke dysphagia with behavioral swallowing interventions is crucial, as dysphagia is associated with increased risk of morbidity and mortality due to aspiration and pneumonia, but also malnutrition and dehydration irrespective of aspiration. In this view, Toscano and colleagues [54] evaluated post-stroke dysphagia upon admission and after 14 days among 275 consecutive patients and revealed that apart from advanced age, lesion size, pre-stroke cognitive impairment and NIHSS $\geq 12$ upon admission, LA severity was independently correlated with early persistent dysphagia.

Similarly, Fandler and colleagues [31,56] focusing on recent small subcortical infarcts, exhibited that early post-stroke dysphagia was associated with higher NIHSS scores on admission, pontine infarcts, presence of microbleeds and, especially, severe baseline LA. The authors propose that post-stroke dysphagia should be considered as a kind of executive dysfunction due to the combined effects of acute focal damage and pre-existent destruction of the subcortical white matter connecting pathways that lower the threshold of input to the medullary swallowing center. Expanding the aforementioned assumptions, Ko and colleagues [57] having studied patients with a first episode of acute unilateral corona radiata (CR) infarction revealed that contralateral corticobulbar tract involvement (CBT), but not total Fazekas scale score, was the independent predictor of post-stroke dysphagia. The CBT constitutes an important tract to consider, as its injury often results in bulbar symptoms, including dysphagia. Thus, the authors suggest that in clinical practice careful interpretation of the localization of LA on brain MRI, especially affecting CBT is crucial to identify patients at increased risk of post-stroke dysphagia.

### 4.4. Short-Term Stroke Outcome (<3 Months)

Neuroimaging markers of SVD encompass lacunes, CMBs, WMHs, and EPVSs. Ryu and colleagues [71] examined the single and cumulative effect of the aforementioned markers, defined as total SVD score, on 3-month functional outcome of AIS patients and exhibited that lacunes, CMBs and WMHs were associated with mRS scores at 3 months. Moreover, the impact of each SVD marker on stroke outcome was smaller than that of the total SVD score. Patients with higher total SVD scores were older, had a history of hypertension, hyperlipidemia and prior stroke. Attempting to explain their findings, the authors proposed that extensive WMH renders the brain more vulnerable to ischemic injury and expansion of ischemic stroke. Moreover, patients with severe WMHs or lacunes may be physically inactive and cognitively impaired, which hampers active rehabilitation and functional regain after ischemic stroke.

Farag and colleagues [73] explored whether LA would impact stroke severity at presentation and short-term clinical. Despite that patients with and without LA did not differ in stroke severity on admission, yet the LA group had a worse outcome at 3 months follow up. This finding is consistent with the theory of compromised brain reserve in the case of LA. Moreover, the authors exhibited that LA degree was positively associated with age, history of hypertension or DM and uric acid levels. High uric acid levels are correlated with inflammation, platelet and endothelial dysfunction. Furthermore, pre-stroke intake of antiplatelets or statins did not prevent the incidence of recurrent strokes or TIAs in LA patients, thus researchers propose that identification of a specific modality of secondary prevention in patients with LA other than that used for territorial strokes is deemed necessary.

To test whether minor cerebrovascular outcome is negatively impacted by pre-existing LA, Zerna and colleagues [67] enrolled 412 patients with high-risk TIA (defined as having either motor or speech deficits lasting ≥55 min) or minor AIS (defined as having baseline NIHSS ≤ 3). After assessing the extent of LA both quantitatively utilizing WML volumetry and qualitatively using Fazekas score, the authors revealed that both higher WML volume and higher Fazekas score were associated with functional disability at 90 days, but they were not associated with stroke progression, TIA recurrence, or stroke recurrence. Moreover, they exhibited that patients with high WML burden have a poorer outcome if they also present with an intracranial occlusion/stenosis. In contrast, pre-existing hypertension, congestive heart failure, atrial fibrillation, DM, smoking and aspirin, Plavix, Aggrenox or warfarin use did not influence short-term functional outcome. Onteddu and colleagues [62] sought to compare chronological age and preexisting LA, an imaging marker of brain frailty and biological age, as prognostic markers of early functional recovery after minor ischemic stroke (defined as NIHSS ≤ 5), as well. Even though both studied parameters were correlated with early functional outcome, after adjustments worse LA, but not advanced chronological age, was independently associated with recovery by 90 days. Similarly, Chen

and colleagues [75] investigated whether neuroimaging markers of SVD, such as WMHs, lacunes, CMBs, EPVSs and the composed total SVD burden score, are associated with short-term outcomes of minor cerebrovascular events (defined as having baseline NIHSS ≤ 3) and demonstrated that only WMH score at baseline was associated with poor functional outcome at 90 days. Researchers also exhibited that early functional outcome is negatively impacted by higher age, NIHSS upon admission and lipoprotein-associated phospholipase A2. However, WMH remained independently associated with less functional recovery after controlling for age and other confounding factors. The authors propose that WMH burden may represent a certain degree of brain fragility, and its addition to chronological age determines an individual's capacity of post-stroke recovery more accurately than chronological age itself and conclude that taking early detection and treatment of WMH into consideration may be beneficial for improving the functional outcome of patients after minor cerebrovascular events.

Ryu and colleagues [64] classified strokes as large artery atherosclerosis, small vessel occlusion, or cardioembolism and investigated how WMH volumes affect NIHSS on admission, END and short-term neurological outcome. The authors revealed that in small vessel occlusion stroke only, higher WMH volume was associated with a higher initial NIHSS score independently of age and infarct volume, indicating a relatively strong influence of WMH on the initial manifestation of this stroke subtype. Moreover, only in large artery atherosclerosis stroke, WMH volume was associated with END. Furthermore, with respect to 3-month functional outcome the stroke subtype that was affected the most by increased WMH seemed to be large artery atherosclerosis stroke, where higher WMH volumes, probably due to more frequent END and worse late recovery. In small vessel occlusion stroke, higher WMH volumes were associated with higher 3-month modified Rankin Scale scores, probably due to more severe initial neurological severity and worse late recovery. In cardioembolism stroke, higher WMH volumes were weakly associated with higher 3-month modified Rankin Scale scores. Finally, in no stroke subtype WMH volume was associated with early or late stroke recurrence.

It may not be too surprising that most patients with small subcortical infarcts (SSI), formerly called lacunar infarcts, have concomitant WMH as both are related to SVD pathology and share common vascular risk factors such as hypertension, dyslipidemia, DM, and tobacco smoking. Even though WMH has been associated with greater ischemic lesion size and worse outcomes after large arterial occlusion, there is a paucity of data regarding the potential association of pre-existing WMH lesion burden with SSI volume and functional outcome. Helenius and colleagues [63] shed light on this matter revealing that apart from SSI volume being related with worse early functional outcome, greater WMH burden was independently associated with larger SSI infarct volumes as well as a poor 90-day outcome after an SSI. Thus, the authors conclude that WMH may represent an easily accessible imaging marker of overall brain health that should be considered as a covariate in future analyses investigating functional outcome after SSI.

Even though symptomatic carotid artery stenosis (CAS) constitutes the major contributor of ischemic stroke, accounting for 10 to 20% of all ischemic strokes, it is still unclear which factors can independently determine the early functional outcome in strokes of the aforementioned pathophysiology. Song and colleagues [76] exhibited that despite the fact that brain-blood flow dynamics (i.e., stenosis degree and flow velocity at the stenotic lesion) and carotid plaque characteristics (i.e., echolucency or calcification) may be risk factors for AIS, neither of them could predict clinical severity or prognosis of AIS. In contrast, larger WMH volume, resembling reduced brain tissue's capacity for recovery, is associated with more severe stroke and poorer prognosis in patients with symptomatic CAS. Researchers propose that based on these results, preventing the WMH risk factors, such as age, male sex, hypertension, history of ischemic stroke and atrial fibrillation may lower the risk of severe stroke and lead to a better prognosis in symptomatic CAS patients. Griessenauer and colleagues [70] compared the impact of WMH grading on 3-month functional outcome of large vessel occlusion (LVO) vs. non-LVO strokes. In both groups, advanced age,

dyslipidemia, anemia, smoking, anemia, COPD, and a family history of stroke, NIHSS scale on admission and increasing WMH volume up to 4 mL were significantly associated with unfavorable functional outcome at 90 days. However, the current study did not demonstrate remarkably different effects of WMH burden on functional outcomes among LVO and non-LVO AIS patients.

With respect to short term outcome, Appleton and colleagues [69] attempted to evaluate how baseline imaging markers of SVD and brain frailty are correlated with clinical outcome after AIS, particularly lacunar stroke. Researchers revealed that the magnitude of SVD and brain frailty was similar in those with lacunar and nonlacunar stroke. Additionally, all measured parameters (LA, cerebral atrophy, or old lacunar infarcts/lacunes) individually, or amalgamated as SVD or brain frailty scores, were associated with unfavorable shifts in mRS score at day 90, with increasing specificity for lacunar stroke.

Despite extensive evaluation, approximately 20 to 25% of AIS patients are diagnosed with cryptogenic stroke (CS), most of them being young. Jeong and colleagues [66] tested whether WMH affect prognosis of CS patients and revealed that severe WMH was independently associated with short-term outcome in young and older CS patients. Old age, female gender, hypertension, smoking, and intracranial and extracranial cerebral artery atherosclerosis were associated with severe WMH. Interestingly, younger CS patients with severe WMH had higher death rates compared to those with no or mild WMH, but the same distinction was not seen in older patients. The authors proposed that the aforementioned association in young CS patients may imply high atherosclerotic burden or hidden cardioembolic sources, such as atrial fibrillation, patent foramen ovale, or atrial septal aneurysm on their case.

Increased supratentorial WMH volume, primarily affecting periventricular white matter and typically sparing the convolutional white matter, U-fibres, corpus callosum, internal capsule and anterior commissure, has been reported as a predictor of worse outcome in AIS patients. Giralt-Steinhauer and colleagues [65] revealed that less common locations, such as brainstem WMHs, independently predicted early stroke outcome, as well. Age, hypertension and DM were significant predictors of supratentorial WMH volume. Brainstem WMH was also associated with age. However, brainstem WMH patients have a slightly different risk factor profile, as hyperlipidaemia and peripheral artery disease (PAD) were clearly associated with brainstem WMH independently of age. Widespread atherosclerotic disease has been linked to greater arterial stiffness and impaired vasoregulatory cerebral mechanisms and according to research, the white matter of the posterior vascular territory is at higher risk.

Sakuta and colleauges [74] having examined 240 non-cardiogenic minor AIS patients (defined as baseline NIHSS score < 4) who were treated with antiplatelet therapy reported that, although high CMB burden was an independent factor for early functional outcome, the same was not proved for deep WMH. However, it should be stated that the authors did not evaluate perivascular WMH which exhibit a closer relationship with brain functional reserve compared to deep WMH. Similarly, Bu and colleagues [61] could not identify a connection between SVD radiological markers, such as WMHs, lacunes, and EPVSs and early stroke outcome. Apart from that, in an attempt to evaluate the predictive value of the "brain before stroke", Coutureau and colleagues [72] examined whether cerebral SVD parameters, such as WMHs, lacunes, EPVSs, CMBs, cerebral atrophy and the total SVD score would add information to the prediction of early stroke outcome compared to validated predictors, such as age and baseline NIHSS. Even though all cerebral SVD parameters were associated with patient outcome they did not provide significant improvement of early stroke outcome prediction. Finally, Schirmer and colleagues [68] revealed that age and systolic blood pressure were significantly positively correlated to WMH and subsequently attempted to improve prediction models of early functional outcome after AIS utilizing structural equation modeling analysis. However, they could also not establish a statistically significant direct association between WMH volume and mRS at day 90.

*4.5. Longterm Stroke Outcome (<1 Year)*

Reid and colleagues [77] aimed to develop models of both excellent (mRS: 0–1) and devastating outcomes (mRS: 5–6) at 6 months post-stroke. Among evaluated radiological variables (any abnormality, any focal abnormality, number of focal lesions, acute infarction, acute infarction or hemorrhage, and LA score), only LA score was associated with either excellent or devastating outcomes. Pre-stroke functional status, ability to lift both arms and stroke severity score (SSS) were predictors of excellent outcomes, whereas age, pre-stroke functional status, normal verbal GCS, total anterior circulation stroke (TACS) and ability to lift both arms were predictors of devastating outcomes. Interestingly, living alone pre-stroke, being a marker of social isolation and associated increased mortality, also predicted higher likelihood of a devastating outcome.

Similarly, Hicks and colleagues [80] revealed that independently of age and lesion characteristics, WMH volume predicted mild to moderate upper extremity motor deficit (defined as having the ability to extend at least 20° at the wrist and 10° at each metacarpophalangeal joint) in patients with post-stroke interval of at least 10 months.

Auriat and colleagues [39] studied the contribution of deep and periventricular WMHs on cognitive and motor performance 6 months post-stroke. The authors exhibited that nonmemory performance was related to periventricular WMH volume, whereas deep WMH volume correlated with both motor function (assessed as the time to complete movements) and motor impairment (indicated by the ability to complete a movement), but was the strongest predictor of motor function.

In 1989, Louis Caplan first used the term branch atheromatous disease (BAD) to describe an occlusion or stenosis at the origin of a deep penetrating artery of the brain, associated with a microatheroma or a junctional plaque, and leading to an internal capsule or pontine small infarct. Liu and colleagues [78] explored whether WMHs and NIHSS scores on admission were associated with 6-month functional outcome in the two main BAD subtypes, i.e., paramedian pontine artery (PPA) and lenticulostriate artery (LSA) atherosclerotic cerebral infarction group. The authors revealed a positive correlation between NIHSS and mRS in both groups, whereas WMHs were associated with 6-month functional outcome only in the PPA group. History of DM, an independent predictor of WMHs, was also more prominent in the PPA group. Compared with anterior circulation, posterior circulation infarcts are closely associated with arteriosclerosis. Thus, the authors suppose that traditional vascular risk factors of arteriosclerosis and WMHs, such as DM, might account for the association between WMHs and functional outcome that was present in the case the PPA region only.

Aphasia is among the most common complications of stroke, whereas the degree of recovery from aphasia is difficult to predict at onset. Wright and colleagues [40] studied the naming outcome of 42 individuals, who initially had aphasia, at least three months after stroke and revealed that LA severity was associated with naming outcome independently of infarct volume, months since onset of stroke, comorbid conditions, and damage to key language areas, such as left inferior frontal gyrus, superior longitudinal fasciculus, and superior temporal gyrus. Apart from aphasia, dysphagia represents one of the commonest complications early post-stroke, as well. Moreover, 13–18% of patients with poststroke dysphagia may experience persistent dysphagia up to 6 months from stroke onset. Lee and colleagues [81] revealed that initial dysphagia severity, bilateral lesions at the corona radiata, basal ganglia, or internal capsule and WMH burden were the main predictors of swallowing difficulties 6 months post stroke, thus exhibiting that the aforementioned clinical and neuroimaging biomarkers are crucial to detect high-risk patients in order to provide sufficient nutritional support and prevent aspiration pneumonia.

WMH usually progress over time. Intriguingly, Wardlaw and colleagues [79] observed that in many minor stroke cases (defined as NIHSS ≤ 7), WMH might conversely regress. Researchers endorsed the idea that prevention of worsening WMH-related brain damage may translate into long-term benefits, as patients with WMH shrinkage had better clinical outcome at 12 months post-stroke, whereas those with WMH increase experienced more

recurrent cerebrovascular events. The authors showed that mean blood pressure (BP) reduction was the main contributing factor to WBH reduction.

### 4.6. Chronic Stroke (>1 Year) Outcome

Oral anticoagulation with either vitamin K antagonists (VKAs) or direct oral anticoagulants (DOACs) constitutes the treatment of choice for secondary prevention of cardioembolic stroke related to atrial fibrillation (AF). However, anticoagulation bears the risk of intracranial hemorrhage. To identify parameters that predispose AF stroke patients treated with anticoagulation to devastating outcomes, Hert and colleagues [84] followed such stroke patients for a median of 2 years and assessed the association of vascular risk factors and MRI features of SVD, such as WMHs and CMBs. The authors revealed that apart from age and concomitant antiplatelet use, both WMHs and CMBs were related to increased risk of recurrent ischemic stroke, intracranial hemorrhage or death within 2 years post-stroke. Moreover, age, WMHs and SVD risk factors, such as DM and hypertension, were independently associated with higher 2 years functional disability. Thus, researchers propose that AF stroke patients with concomitant SVD should be guided thoroughly during the post-stroke phase with intensive control and treatment of existing vascular risk factors to reduce the chances of long-term poor functional outcome.

Renal dysfunction and SVD not only share common underlying pathophysiological mechanisms but are also both associated with poor prognosis after AIS. Jeon and colleagues [83] aimed to investigate the prognostic relationship between renal dysfunction, evaluated with estimated glomerular filtration rates (eGFRs), SVD neuroimaging markers, such as silent lacunar infarction, WMHs and CMBs, and 3 years survival rate post-stroke. The authors revealed that patients with high eGFR and no WML had the best survival rate, whereas three-year survival rate was lower in patients with renal dysfunction and each type of SVD. Moreover, after adjusting for age and eGFR, the presence of WML was the only SVD factor related to mortality.

Patients with large artery atherosclerosis (LAA), defined as significant (50%) stenosis of the large cerebral artery relevant to the acute infarction, frequently have WMHs, as LAA and WMH share common underlying pathophysiological mechanisms, like reduced cerebral perfusion and increased arterial stiffness. Baik and colleagues [82] investigated the impact WMH on short- and long-term outcome of patients with LAA. Hypertension and higher initial NIHSS score, but not WMH burden were the main predictors of poor functional outcome at 3 months. In contrast, in chronic stroke patients, especially in the elderly older age, history of diabetes, serum creatinine level, initial NIHSS score and WMH severity were all independent predictors of all-cause and cardiovascular mortality. The relationship between severe WMH and long-term mortality in patients with LAA remained significant even after adjustment for well-known predictors including age and initial NIHSS score.

### 4.7. Rehabilitation Outcome

An increasing number of stroke survivors needs dedicated rehabilitation. Nevertheless, recovery after intensive rehabilitation remains difficult to predict. Functional Independence Measure (FIM) is widely used for assessment of the degree of independence in activities of daily living (ADL) in poststroke patients. Senda and colleagues [85] showed that age and total FIM score on admission were significant predictors of total FIM score at discharge from rehabilitation. Interestingly, when total FIM score was subdivided into motor and cognitive components, the authors revealed that periventricular WMH grade correlated with motor function, whereas deep WMH grade was associated with cognitive function. Periventricular WMH is defined as hyperintensity continuous with the periventricular region, whereas deep WMH as a lesion in the subcortical region but not continuous with the periventricular region. Injury of the long association and projection fibers that connect the different lobes that are present in large numbers in periventricular regions may explain

reduced motor function, whereas deep WMH injury of the short association fibers, which connect between or within the cerebral lobes, may lead to impaired cognitive function.

Several factors have been identified to impact poststroke disability, including patient age, initial deficit severity, infarct size and location and LA. Khan and colleagues [86] assessed the FIM motor score and FIM cognitive score at discharge of 109 consecutive post-AIS patients admitted to acute inpatient rehabilitation. Researchers found that the degree of preexisting LA is associated with rehabilitation outcomes in a domain-specific fashion. More specifically, the FIM motor score at discharge was not predicted by age, initial NIHSS, infarct volume, or LA severity, whereas infarct volume and LA severity independently predicted lower FIM cognitive scores at discharge. Thus, the authors conclude that rehabilitation programs should focus on cognitive rehabilitation for patients with severe LA.

Recovering gait and balance control is a prerequisite for satisfactory mobility post-AIS. Several factors may affect balance and gait recovery after stroke, including stroke volume and severity and disruption of the corticospinal tract. Dai and colleagues [87] examined whether preexisting LA may also affect balance and gait recovery post-AIS and revealed that along with age, initial NIHSS score, lesion volume and disrupted corticospinal tract, WMH severity was independently associated with poor gait recovery and increased risk of falls at discharge from rehabilitation. In contrast, stroke type, sex, and BMI were not independently associated with poor gait recovery. The authors propose that in routine clinical practice the severity of LA should help establish the prognosis in terms of mobility recovery, thus solutions that ensure patient safety, such as wheelchair use and appropriate house adaptations, could be implemented as soon as possible.

### 4.8. Current Guidelines

In 2021, the European Stroke Organisation (ESO) developed guidelines to assist clinicians with management of SVD, specifically WMHs and lacunes [88]. The authors revealed that SVD patients with concomitant hypertension or DM should have their BP or blood glucose well controlled, respectively. However, there was no evidence to support the use of a specific antihypertensive drug class or specific agent for obtaining appropriate glycemic control or administration of antihypertensive or antiglycemic drugs to SVD patients that do not have hypertension or DM. With regard to hyperlipidemia, evidence on the benefits of lipid lowering to halt WMHs progression is limited; thus, the Guideline Group were divided on whether statins should be considered in SVD. With respect to lifestyle interventions all group members suggested that there is no direct evidence supporting that any specific type of lifestyle intervention might prevent clinical outcomes in patients with SVD. However, healthy diet, smoking cessation, aerobic exercise, and avoidance of obesity are strongly encouraged, although this is for general health reasons rather than specifically because of evidence of benefit in SVD. Interestingly, researchers concluded that conventional antiplatelet drugs, such as aspirin and clopidogrel, should not be prescribed to SVD patients, unless there is some other justifiable reason, such as ischemic heart disease or TIA/minor stroke. This finding is in keeping with the perspective that SVD is not primarily atherothromboembolic; of note, SVD stroke subtype was the only stroke subtype not associated with a genetic risk score for thrombosis based on previous genetic studies. Finally, the Guideline Group did not favor routine use of follow-up neuroimaging to evaluate progression of SVD, as visible SVD features do not reflect the full extent of brain damage and, so far, subvisible markers of SVD have limited acceptance. The authors conclude that future studies should investigate whether interventions should differ according to SVD subtype, since, for example, WMHs and lacunes share different underlying pathologies. Moreover, they suggest that modern genomics and molecular approaches should focus on identifying additional biological pathways involved in SVD generation, thus facilitating drug development specific for SVD.

### 5. Conclusions

Taking everything into consideration, the present review provides an overview of the potential clinical applications of the evaluation of pre-stroke LA as a prognostic biomarker of functional and rehabilitative outcome in an acute ischemic stroke setting. Our systematic review is not without limitations. First, it is possible that different pa-pers published by the same research groups reported findings on the same cohort. Moreover, the mediator and moderator role of sociodemographic characteristics on the association between LA and patients' clinical outcome has not been thoroughly stud-ied. However, we provided a comprehensive presentation of patients' quantitative characteristics (sociodemographic and clinical data) which might enable indirect evaluation of such roles and provide valuable information for the implementation of future studies. Despite the lack of consensus amongst the literature, our findings support the use of LA assessment as a useful prognostic factor in stroke, thus indicating that a biomarker-based approach may provide important insight into the recovery potential of each stroke survivor and the means to establish a prognosis of mobility. This information may aid counseling for patients and families, improve the selection of more appropriate radical therapy with some brain-protecting agents and early rehabilitation, as well as help tailor rehabilitative efforts to fit patients' needs. LA seems to reliably differentiate between patients with good prognosis and patients with an unfavorable functional outcome from early up to chronic follow up and discern those more prone to hemorrhagic transformation, probably because patients with pre-existing brain damage are less able to show functional compensation than those without pre-existing damage. Given that the degree of LA can add valuable predictive information beyond clinical or other neuroimaging variables, it could significantly enhance the discriminatory accuracy of widely utilized validated prognostic scores, thus optimizing the overall stroke management, and facilitating individualized stroke care. Additional studies among stroke patients on the association between LA and inclination for recovery are recommended to further illuminate this clinically important relationship. Moreover, future meta-analysis and mega-analysis studies may also examine this association, further attempting to reach to conclusions which may assist to the formation of guidelines for the best course management for the AIS patients in several settings based on the pre-existing LA.

**Author Contributions:** A.S. and D.T. reviewed the literature, screened the abstracts of the reference list, deleted duplicates and citations not meeting the inclusion criteria, and assessed the articles; K.V. solved any disagreement regarding screening, or selection process; F.C. wrote the first manuscript; A.S., S.A., E.A.P., N.A., E.K. and K.V. reviewed the tables, the presentation of the data, and the methodology. The corrected version was discussed collegially. F.C., C.K., S.K. (Stefanos Karamanidis), K.T., S.K. (Sofia Kitmeridou) and S.K. (Stella Karatzetzou) wrote the final version. All authors have read and agreed to the published version of the manuscript.

**Funding:** We acknowledge support of this work by the project "Study of the interrelationships between neuroimaging, neurophysiological and biomechanical biomarkers in stroke rehabilita-tion (NEURO-BIO-MECH in stroke rehab)" (MIS 5047286) which is implemented under the Ac-tion "Support for Regional Excellence", funded by the Operational Program "Competitiveness, Entrepreneurship and Innovation" (NSRFm2014-2020) and co-financed by Greece and the European Union (European Regional Development Fund).

**Institutional Review Board Statement:** Not applicable.

**Informed Consent Statement:** Not applicable.

**Data Availability Statement:** All data discussed within this manuscript is available on PubMed.

**Conflicts of Interest:** The authors declare no conflict of interest.

## Abbreviations

AIS: Acute ischemic stroke; ARWMC: Age-related white matter change; BAD: Branch atheromatous disease; BI: Barthel index; CBT: Contralateral corticobulbar; CHS: Cardiovascular Heart Study; CMB: Cerebral microblees; CS: Cryptogenic stroke; CSVD: Cerebral small vessel disease; cSVD: Cerebral small vessel disease; CT: Computed Tomography; DWMH: Deep white matter hyper intensity; eGFR: estimated Glomerular Filtration Rate; END: Early neurological deterioration ENI: Early neurological improvement; FIM: Functional Independence Measure; FLAIR: Fluid-attenuated inversion recovery; GCS: Glasgow coma scale; HT: Hemorrhagic transformation; LA: Leukoaraiosis; LAA: Large artery atherosclerosis; MBI: Modified Barthel Index score; MCA: Middle cerebral artery; MMSE: Mini mental state examination; mRS: Modified Ranking score; NIHSS: National Institutes of Health Stroke Scale; NM: Not mentioned; PPA group: Paramedianpontine artery group; PRISMA: Preferred reporting items for Systematic Reviews and Meta-analyses; PVH: Periventricular Hyperintensity; PVWMH: Periventricular white matter hyperintensities; SVD: Small vessel disease; TIA: Transient ischemic attack; TMS: Transcranial Magnetic Stimulation; TSI: Transient symptoms with infarction; VSS: Van Swieten Scale; WAB: Western Aphasia Battery; WM: white matter; WMH: White Matter Hyperintensity; WML: White matter lesion.

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
