# Peer review of "The Clinical Utility of Leukoaraiosis as a Prognostic Indicator in Ischemic Stroke Patients"

_2035-8377, doi:10.3390/neurolint14040076_

Round 1

Reviewer 1 Report

This is a very interesting and original review on a relevant topic.

However some major points should be carefully clarified prior to  publication. 
The introduction seems to be rather sparse and a better and more focused and complete overview of leucaraiosis is mandatory. In this view authors should include the concept of leucaraiosis, epidemiological data, its clinical and wide presentation, the role of neurochemical and neurophysiological mechanisms recently identified, neurophatology data, data from structural and functional magnetic resonance imaging, classification, the LADIS studies with the prognostic values, the intruiguing data about cognitive impairment and mood disorders related to leucaraiosis (vascular dementia and depression), the link with heart diseases, the concept of microbleeds...

Material and methods (data extraction)
for every article authors should extract also the patients demographic and clinical characteristics (age, sex, the level of education, marital/occupational status, height and weight, cardio and cerebrovascular risk factors such as hypertension, atrial fibrillation, coronaropathy, drugs previous stroke) and insert in the tables.

Results:
Please include 2 table according to time of clinical outcome (early vs late).

Discussion:
this section is sometimes confusing. I suggest to globally revise it, including the possible pathophysiological mechanism linking leukaraiosis and the different prognosistic measure of interest. This aspect requires careful attentions and needs to be discussed (consider all the variables such as age, hypertension at admission, sex, previous treatments, other metabolic diseases, secondary treatment for prevention).  Finally what about guidelines suggest? Authors should include a graphical model to propose the relevance of the "leukaraiosis factor" in choosing the best course of management for the patient with acute stroke in the several settings.

Finally authors should submit the manuscript to a linguistic revision by a native English speaker.

Author Response

Response to Reviewer

Comments

Reviewer: This is a very interesting and original review on a relevant topic. However some major points should be carefully clarified prior to  publication. 

Response: We thank the reviewer for dedicating time to review this manuscript and the encouraging comments. We have carefully considered each suggestion and have thoroughly revised our initial submission. All changes are highlighted in the revised text and table.

Point 1: The introduction seems to be rather sparse and a better and more focused and complete overview of leucaraiosis is mandatory. In this view authors should include the concept of leucaraiosis, epidemiological data, its clinical and wide presentation, the role of neurochemical and neurophysiological mechanisms recently identified, neurophatology data, data from structural and functional magnetic resonance imaging, classification, the LADIS studies with the prognostic values, the intruiguing data about cognitive impairment and mood disorders related to leucaraiosis (vascular dementia and depression), the link with heart diseases, the concept of microbleeds...

Response 1: We appreciate the reviewer’s suggestions for the introduction section. We carefully considered all the suggestions and modified the introduction accordingly.

Point 2: Material and methods (data extraction). For every article authors should extract also the patients demographic and clinical characteristics (age, sex, the level of education, marital/occupational status, height and weight, cardio and cerebrovascular risk factors such as hypertension, atrial fibrillation, coronaropathy, drugs previous stroke) and insert in the tables.

Response 2: We thank the review for the suggestion. Apart from participants’ age which was included, we included sex and level of education (when available). We also carefully searched each article for additional clinical characteristics and included all available data. Data extraction and tables have been modified accordingly.

Point 3: Results: Please include 2 table according to time of clinical outcome (early vs late).

Response 3: We really appreciate the reviewer’s comment regarding early and late clinical outcome. Considering that Table 1 is already extensive, we provided additional details regarding early and late outcome by adding a new paragraph in the Results section.

Point 4: Discussion: this section is sometimes confusing. I suggest to globally revise it, including the possible pathophysiological mechanism linking leukaraiosis and the different prognosistic measure of interest. This aspect requires careful attentions and needs to be discussed (consider all the variables such as age, hypertension at admission, sex, previous treatments, other metabolic diseases, secondary treatment for prevention).  Finally what about guidelines suggest? Authors should include a graphical model to propose the relevance of the "leukaraiosis factor" in choosing the best course of management for the patient with acute stroke in the several settings.

Response 4: We thank the reviewer for raising these important points. We have extensively revised the Discussion section accordingly. Regarding the graphical model of guidelines, we carefully considered the suggestion and discussed with our group. Considering the lack of consensus among the literature and specifically the heterogeneity of the studies with regards to several methodological issues (which are now presented in detail in Results section, Table 1 and Discussion section after the reviewer’s constructive suggestions), we believe that such a graphical model could not be informative, since an attempt to simply “translate” the current main findings into guidelines for the course management of AIS based on leukoaraiosis might negatively affect the validity of the graphical model. However, we do believe that future meta-analysis and mega-analysis studies might address this issue and provide quite valid guidelines. We have included this suggestion in our revised Discussion section. Thank you.

Point 5: Finally authors should submit the manuscript to a linguistic revision by a native English speaker.

Response 5: According to the reviewer’s suggestion, the revised manuscript was carefully checked and edited for the English language by a native English speaker.

Reviewer 2 Report

The authors produced a comprehensive review of articles linking leuckoareussi with stoke in the last decade. relevant studies were divided into five groups and briefly described. The article is comprehensive and well written, albeit rather dry and the conclusions are not novel

Author Response

Dear Reviewer,

Many thanks for your kind words.

The whole manuscript wassignificantly upgraded to avoid being"dry"

Yours Sincerely

Dr Tsiptsios

Round 2

Reviewer 1 Report

The authors responded very carefully and comprehensively to the previous comments.

I invite the authors to check the references in the manuscript.

Author Response

We thank the reviewer for dedicating time to review this revised version of our manuscript. We do acknowledge that a substantial re-arrangement of the text was necessary to address reviewer’s previous comment in order to improve the final manuscript. We have carefully checked all the references both in text, tables, and reference list (both manually and using EndNote) and we verify that the numerical order of all references is correct. Thank you very much for addressing this point.